# Helical-caging enables single-emitted large asymmetric full-color circularly polarized luminescence

Yajie Zhou[1], Yaxin Wang[1], Yonghui Song[2,3], Shanshan Zhao[1], Mingjiang Zhang[1], Guangen Li[1], Qi Guo[1], Zhi Tong[1], Zeyi Li[1], Shan Jin[4,5,6], Hong-Bin Yao [2,3], Manzhou Zhu [4,5,6] & Taotao Zhuang [1,2] ✉

Colorful circularly polarized luminescence materials are desired for 3D displays, information security and asymmetric synthesis, in which single-emitted materials are ideal owing to self-absorption avoidance, evenly entire-visible-spectrum-covered photon emission and facile device fabrication. However, restricted by the synthesis of chiral broad-luminescent emitters, the realization and application of high-performing single-emitted full-color circularly polarized luminescence is in its infancy. Here, we disclose a single-emitted full-color circularly polarized luminescence system (spiral full-color emission generator), composed of whole-vis-spectrum emissive quantum dots and chiral liquid crystals. The system achieves a maximum luminescence dissymmetry factor of 0.8 and remains an order of $10^{-1}$ in visible region by tuning its photonic bandgap. We then expand it to a series of desired customized-color circularly polarized luminescence, build chiral devices and further demonstrate the working scenario in the photoinduced enantioselective polymerization. This work contributes to the design and synthesis of efficient chiroptical materials, device fabrication and photoinduced asymmetric synthesis.

Circularly polarized luminescence (CPL) has aroused universal concern these years owing to its versatile utilization and application potential in anti-counterfeiting[1–3], 3D optical displays[4–6], photoelectronic devices[7–9], disease detection[10], asymmetric synthetic photochemistry[11] and the promotion of plant growth[12]. However, most CPL employed in practical applications is produced by lasers, polarizers and quarter-wave plates—not conducive to the miniaturization of optical systems and the construction of wearable devices. Additionally, for the small luminescence dissymmetry factor ($g_{lum}$), practical applications based on CPL-active materials have been slow to get off the ground for a long time, so searching for high quality materials with large $g_{lum}$ values has always been the key towards practical applications in this field. Among all CPL, single-emitted full-color circularly polarized luminescence (SF-CPL) must be taken care of since it can be a considerable candidate for cryptography[13,14], affecting the biomass production of plants[12] and the accurate production of customized-color CPL[15].

One single emitter with broad-vis-spectrum luminescence is of particular importance all along since it allows to avoid energy transfer among mixed components and simplify device fabrication[16–18], but to

[1]Department of Chemistry, University of Science and Technology of China, Hefei 230026, PR China. [2]Hefei National Research Center for Physical Sciences at the Microscale, University of Science and Technology of China, Hefei 230026, PR China. [3]Department of Applied Chemistry, University of Science and Technology of China, Hefei 230026, PR China. [4]Department of Chemistry and Centre for Atomic Engineering of Advanced Materials, Anhui University, Hefei 230601, PR China. [5]Key Laboratory of Structure and Functional Regulation of Hybrid Materials of Ministry of Education, Anhui University, Hefei 230601, PR China. [6]Key Laboratory of Functional Inorganic Material Chemistry of Anhui Province, Anhui University, Hefei 230601, PR China. ✉e-mail: tzhuang@ustc.edu.cn

date, materials with single-emitted full-color CPL are still absent. Intensive efforts have produced chiral organic molecules[19–37], MOFs[38,39], polymers[40–46] and inorganic nanostructures[47–49] with multi-color emission. Nonetheless, these full-color-CPL-active materials still cannot satisfy accurate practical requirements because of the complicated and imprecise regulation of multi-emitters, so it is pressing and of particular importance to develop SF-CPL-active materials with large $g_{lum}$ values, simplified adjustment and customized potential.

To achieve high-performing SF-CPL, we took a view, combined high-quality inorganic white emitters with a large asymmetric chiral host[5,24,25,50–52]. In this work, we describe a SF-CPL system (spiral full-color emission generator, SFEG), offering the visible-spectrum-regulated CPL with $g_{lum}$ values up to 0.8. Since the shelling process can adjust the luminescent properties and improve the photoluminescence quantum yield (PLQY) of quantum dots (QDs)[17,53], we chose the core-shell QDs as the highly-efficient emitter. The prepared white luminescent Cu-Ga-S core was wrapped in double ZnS shells to enhance the PLQY to 66% and tune the photon emission to the whole visible range. We then controlled the photonic bandgap of the chiral nematic liquid crystal to achieve full-color CPL and further extended to white CPL based on the liquid crystal polymer. The continuous wide-spectrum-emission of the SFEG also made it possible to obtain customized CPL for device fabrication and further practice. Here, we also successfully employed the highly-efficient SFEG to co-induce the enantioselective polymerization of 10,12-tricosadiynoic acid, providing more practicalities for the application of CPL in photon-driven asymmetric synthesis.

## Results and discussion
### Spiral full-color emission generator design
The generation of SF-CPL depends largely on the precise synthesis of the white-luminescent emitter, the selection of CPL-amplifying photonic crystal and the ingenious combination of these two. Here, we design a long-range ordered cage-like assembly strategy: the white quantum dots (WQDs) are wrapped in the chiral nematic liquid crystal

similar to a night pearl in a spiral cage—converting natural white light to large asymmetric full-color CPL.

It is of high advantage to choose a single QD with broad-vis-range emission instead of mixed ones to achieve white luminescence because of simplified construction of photoelectric device as well as stable emission in practical applications. Achieving complete visible spectrum emission from one single kind of QD can be proceeded by doping impurity ions such as $Mn^{2+}$ or $Cu^+$ into the QD host[54,55]. Here, with the developed multiple-step injection method[17], we synthesized the WQDs with a high PLQY of up to 66%. Figure 1 illustrates the construction of the SFEG: the Cu-doped Ga-S core is first prepared by a heat-up approach, and the corresponding multiple ZnS shell counterparts are wrapped over the Cu-Ga-S core by a two-step hot injection method (Supplementary Fig. 1). The synthesized Cu-Ga-S/ZnS core-shell QDs show broad-vis-region white luminescence. Then, the nematic liquid crystal 5CB is selected as a suitable chiral host for its macroscopically favorable compatibility, room-temperature stability, regular fingerprint textures and periodic spiral arrangement induced by chiral dopants (e.g., R/S811, offering large helical twisting power). When combined to form the CPL-amplifying soft photonic crystal, the system effectively boosts the CPL signals throughout. Meanwhile, thanks to the broad luminescence range of the emitter, the chiral nematic liquid crystal's photonic bandgap could be tuned in the vis-spectrum without the change of $g_{lum}$ values' order of magnitude. With the ordered integration of the high-quality WQDs and the appropriate chiral nematic liquid crystal, we achieve the considerable SFEG system.

### Characterizations of the spiral full-color emission generator
We first used transmission electron microscopy (TEM) to examine the morphology of the resultant WQDs (Fig. 2a, Supplementary Fig. 2) and further confirmed the crystallization of the QDs as well as observing their size is concentrated at ca. 3 nm by high resolution transmission electron microscopy (HRTEM) (Fig. 2b), which indicates the potential to realize high-quality photoluminescence (PL). We performed energy-dispersive X-ray spectroscopy (EDS) mapping analysis (Supplementary Fig. 3) to evidence the successful doping of $Cu^+$ in the WQDs. The X-ray

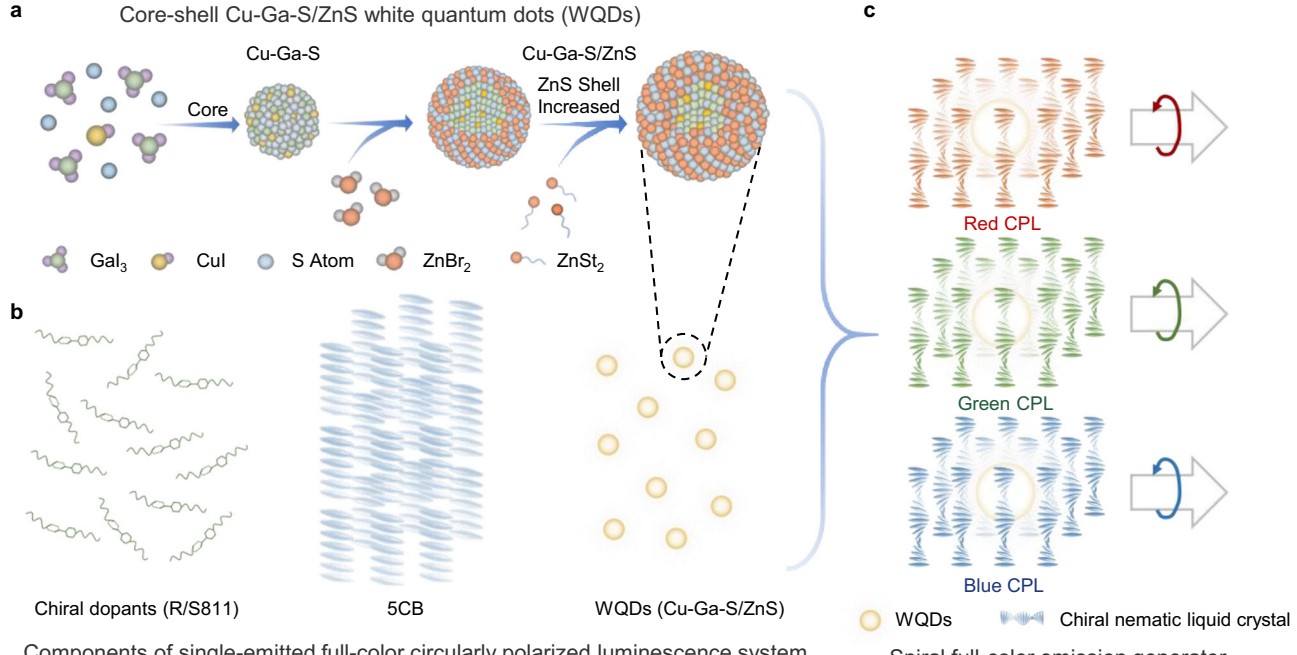

**Fig. 1 | Schematic illustration of the fabrication of the spiral full-color emission generator (SFEG). a** A multiple-step injection method for the synthesis of core-shell Cu-Ga-S/ZnS white quantum dots (WQDs). **b** Components of the SFEG. **c** Schematic drawing of the SFEG formed by a long-range ordered cage-like assembly strategy.

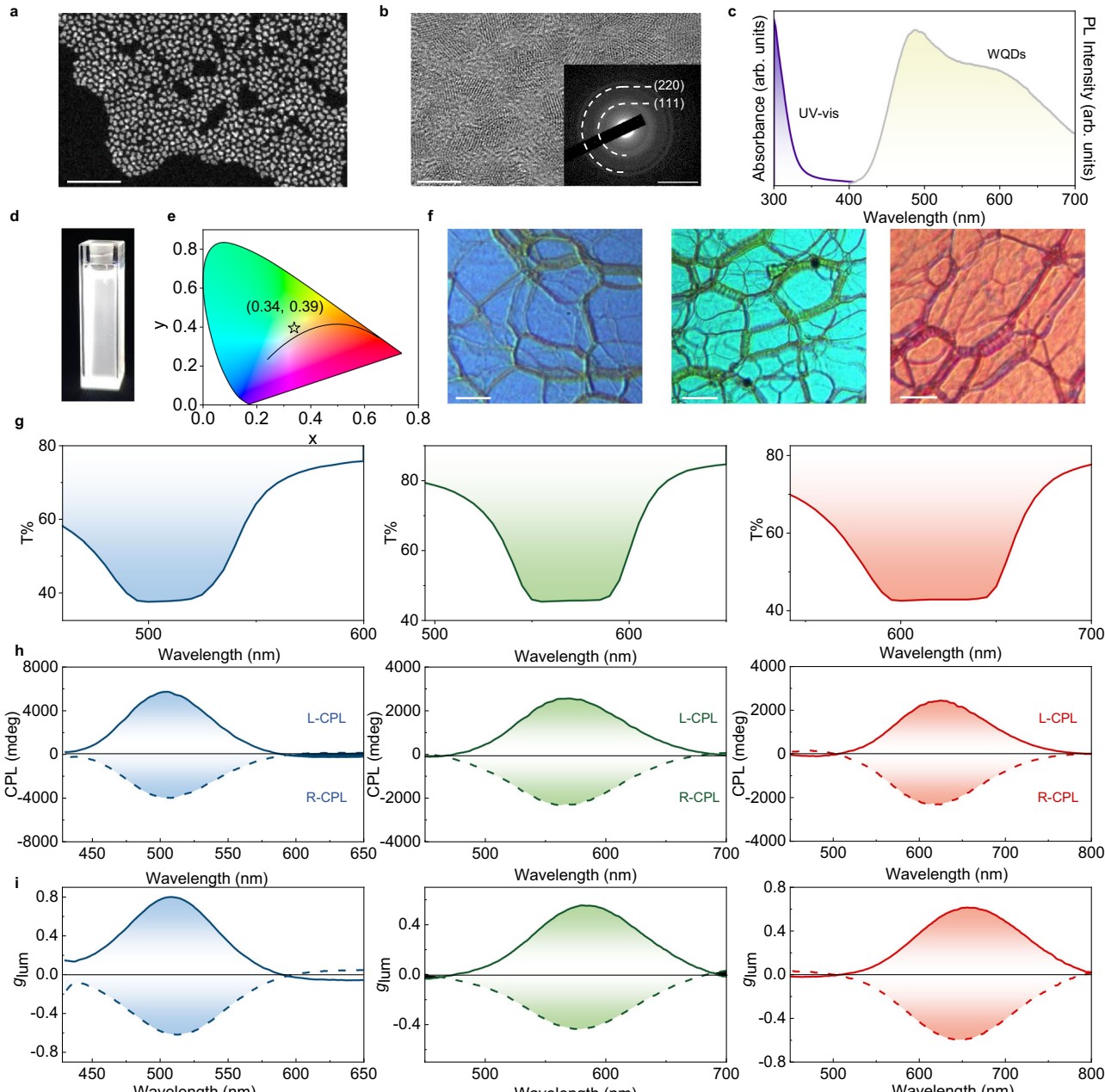

**Fig. 2 | Characterizations of the Cu-Ga-S/ZnS white quantum dots (WQDs) and spiral full-color emission generator (SFEG). a** High-angle annular dark field scanning transmission electron microscopy (HAADF-STEM) image of the WQDs. Scale bar = 50 nm. **b** Representative high resolution transmission electron microscopy (HRTEM) image of the WQDs (scale bar = 5 nm) with inset corresponding to the selected area electron diffraction (SAED) pattern (scale bar = 5 1/nm). **c** Absorbance (UV-vis) and photoluminescence (PL) spectra of the WQDs in toluene. **d** Fluorescent image of the WQDs in a quartz cell with toluene as the solvent. The bottom area of quartz cuvette is 1 cm². **e** The corresponding CIE color coordinates of the WQDs. **f** Polarizing optical microscopy (POM) images of the representative SFEG. Scale bar = 100 μm. **g-i** Transmission spectra (**g**), circularly polarized luminescence (CPL) spectra (**h**) and the corresponding luminescence dissymmetry factor ($g_{lum}$) values (**i**) of the representative SFEG. Source data are provided as a Source Data file.

diffraction (XRD) pattern showed the archetypal tetragonal chalcopyrite structure of the WQDs (Supplementary Fig. 4).

We then adopt various characterization instruments to study the photochemical properties of the SFEG. The absorbance and PL spectra of the WQDs are shown in Fig. 2c, demonstrating that the intense absorption band is located in the ultraviolet (UV) region, and the emission covers the visible range from blue-purple to red under UV irradiation. Such broad spectrum enables the WQDs to emit bright white luminescence (Fig. 2d), with the corresponding CIE color coordinates measured as (0.34, 0.39) (Fig. 2e). We thereafter doped these synthesized WQDs (*i.e.*, single-broad-luminescent emitter) into the designed chiral nematic liquid crystal host (5CB-R/S811) with optimized reaction conditions (details in Supplementary methods). We tuned the ratio of chiral dopants and 5CB in the hybrid system—obtaining different pitches and distinctive structural colors (Supplementary Figs. 5–7)—to achieve full-color CPL (Supplementary Figs. 8–11). All fabricated SFEG (for example, doped with 29 wt%, 26 wt% and 23 wt% R/S811) showed typical fingerprint textures (Fig. 2f), proving the formation of chiral structures. As expected, obvious photonic bandgaps (Fig. 2g) and strong symmetric CPL signals were detected (Fig. 2h, the density of states oscillations in the edge of the photonic bandgap leading to the weak CPL reverse signals[56]) with the

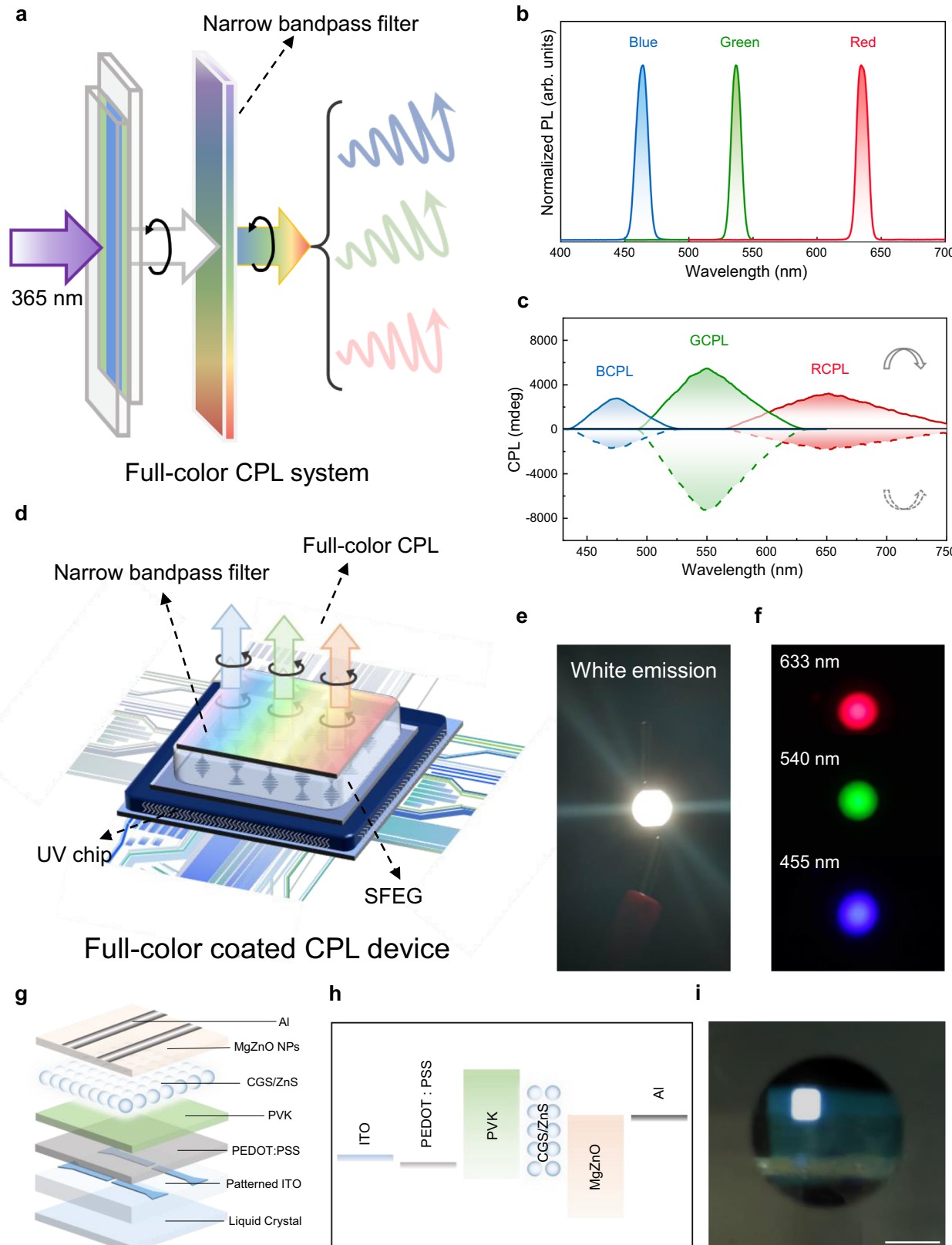

**Fig. 3 | Schematic illustration showing the visualized, customized-color circularly polarized luminescence (CPL) and CPL device fabrication. a** Schematic drawing of the customized-color CPL systems by covering customized-color narrow bandpass filters over the spiral full-color emission generator (SFEG). **b-c** Photoluminescence (PL) spectra (**b**) and the corresponding mirror-imaged CPL spectra (**c**) of the visualized customized-color CPL system. Source data are provided as a Source Data file. **d** The structure diagram of the proof-of-concept customized-color coated CPL devices by covering customized-color narrow bandpass filters over a coated CPL device. **e-f** Photographs of the white-emission (**e**) and customized-color coated CPL devices (**f**). **g** Schematic of the electroluminescent CPL device. **h** Energy band diagram of a solution-processed, bottom-emitting multilayered white QLED. Indium-Tin Oxide (ITO), poly(3,4-ethylenedioxythiophene):poly(styrenesulfonate) (PEDOT:PSS), polyvinylcarbazole (PVK), white quantum dots (CGS/ZnS). **i** Bright image of the electroluminescent CPL device. Scale bar = 5 mm.

$g_{lum}$ value up to 0.8 (Fig. 2i). We also extended to white CPL with the help of the broad-photonic-bandgap liquid crystal polymer (Supplementary Figs. 12, 13).

## Visualized customized-color circularly polarized luminescence and device construction

Most full-color CPL systems were realized by mixing monochromatic emitters in certain proportions[23,27,28,32–34,37,39,49,57]; unfortunately, it is particularly complicated to obtain the required CPL with a specific wavelength while maintaining color stability, and such a strategy increased the complexity of the device construction—not friendly for practical applications. The SFEG we elaborated avoids these issues but shows full-color CPL signals under the veil of bright white emission. We sought, therefore, to cover our SFEG using different narrow bandpass filters (Fig. 3a) that would enable to obtain visualized CPL with accurate emission in a facile way and ultimately realize coated CPL device construction. As a result, various customized-color emissions with satisfying full-width at half-maximum were produced, as exhibited in Fig. 3b. Meanwhile, we achieve strong mirror-image CPL signals and large $g_{lum}$ values with corresponding emissions (Fig. 3c, Supplementary Fig. 14), which provides a method to form desired CPL for photoelectronic and next-generation 3D display applications.

On account of the delightful customized-color CPL performance, we built a prototype coated white-emission CPL device (Fig. 3d)—coating the as-prepared SFEG on the surface of a commercially available UV-LED chip. The achieved coated CPL device showed bright white emission (Fig. 3e). After combining different narrow bandpass filters, we also constructed customized-color coated CPL devices with accurate red (633 nm), green (540 nm) and blue (455 nm) emissions (Fig. 3f), offering a thread to simplify the display fabrication. Furthermore, we built an electroluminescent CPL device on the basis of an improved liquid crystal system[17,58]—adding the highly asymmetric freestanding liquid crystal film bottom on the multilayered white QLED (Fig. 3g–i, Supplementary Fig. 15), which makes a significant step towards the construction of large asymmetric CPL devices.

## Circularly polarized luminescence-induced enantioselective polymerization of 10,12-tricosadiynoic acid

The successfully prepared SFEG with strong CPL signals encourages us to put it into real applications. We showed a photon-to-matter case, that is, employing our SFEG to generate CPL and then illuminating the light on the 10,12-tricosadiynoic acid (TDA) monomer films for growing the chiral poly 10,12-tricosadiynoic scid (PTDA) (Fig. 4a, Supplementary Figs. 16, 17). After simultaneous explosion under left- or right-handed CPL (Fig. 4b) and shortwave UV for 15 s, the virtually transparent TDA films turned to blue and the UV-vis and CD analysis of the samples showed obvious absorption and opposite Cotton effect, proving the formation of the chiral PTDA (Fig. 4c, d). We also conducted other comparative experiments to confirm the interaction between CPL and TDA (Supplementary Figs. 18–24, Supplementary Table 1), evidencing that the polymerization relies on shortwave UV while the enantioselectivity depends on the handedness of CPL. The photon-driven chirality transfer using the fabricated SFEG device, enables to promote the understanding of chirality origin in nature and the induction of optical, electrical and magnetic properties to chiral intelligent materials[10,59–62].

Last but not least, since smart CPL materials responsive to external stimulation have attracted enormous attention for the demand of information storage, encryption and anticounterfeiting[1–3,63,64], the as-prepared SFEG could also be a potential candidate for intelligent CPL anti-counterfeiting due to its sensitive response to ambient temperature and UV light (Supplementary Figs. 25, 26), further broadening the application fields of CPL.

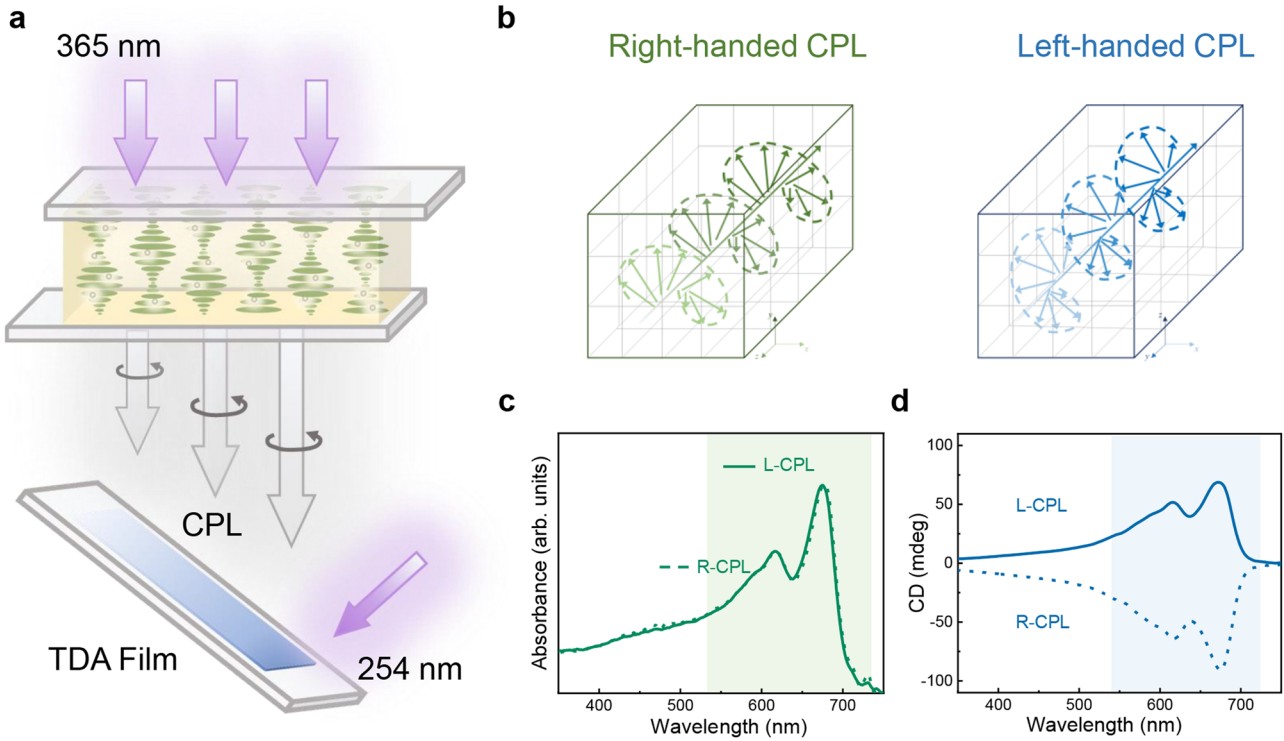

**Fig. 4 | Enantioselective polymerization of 10,12-tricosadiynoic acid (TDA) using the spiral full-color emission generator (SFEG). a** Schematic drawing of the enantioselective polymerization of TDA induced by circularly polarized luminescence (CPL) from the SFEG (peak at 500 nm, $|g_{lum}| = 0.6$) assisted by shortwave UV (UV lamp: 254 nm, 16 W; the sample-light distance: 15 cm). **b** Two different illuminant conditions: left- (blue) and right-handed (green) CPL. Absorbance (**c**) and circular dichroism spectra (**d**) of the chiral poly 10,12-tricosadiynoic acid (PTDA) induced by left- (solid curve) and right-handed (dashed curve) CPL. Source data are provided as a Source Data file.

In summary, we present a spiral full-color emission generator (SFEG) with a largest $g_{lum}$ value of 0.8. We then achieve customized-color CPL, providing a simplified approach to obtain precise CPL, and after that, the coated photoluminescent and electroluminescent CPL devices are constructed to make preliminary attempts for practice. Furthermore, the as-prepared SFEG is employed as a direct CPL generator in the photoinduced enantioselective polymerization of TDA—showing opposite CD signals under the illumination of left- or right-handed CPL assisted with shortwave UV. This work paves the way for the synthesis and practical application of CPL in photo-induced asymmetric synthesis, device construction and even future displays.

## Methods

### Synthesis of the white quantum dots

For the synthesis of the Cu-Ga-S cores[17], 5.95 mg Cu iodide, 112.6 mg Ga iodide, 0.25 mL dodecanethiol (DDT) and 2.5 mL oleylamine (OLA) were put in a 50 mL three-neck flask, and then heated to 100 °C with degassing and $N_2$-purging procedures back and forth three times. After that, the mixture was heated to 175 °C, and the sulfur stock solution (dissolving 32 mg S powder in 1 mL 1-octadecene at 180 °C) was injected into the mixture in the three-neck flask at 175 °C for 5 min. For the synthesis of the consecutive ZnS shells, two different ZnS stock solutions were prepared by dissolving 366 mg Zn acetate in a mixture of 1 mL DDT, 1 mL ODE and 2 mL oleic acid and 2.53 g Zn stearate in a mixture of 2 mL DDT and 4 mL ODE, respectively. The first ZnS stock solution was heated to 120 °C and injected into the Cu-Ga-S core solution, and then the reaction was heated to 210 °C for 30 min. Thereafter, the second ZnS stock solution was injected and the reaction was kept at 245 °C for 60 min. The growth solution was cooled to 80 °C and diluted with toluene, and then washed three more times using toluene/ethanol mixture by centrifugation at $15,777 \times g$ for 10 min and finally redispersed in toluene.

### Fabrication of the spiral full-color emission generator

For the fabrication of the representative blue SFEG, 1 mL the as-prepared WQD toluene solution, 0.29 g R/S811 were mixed well, stirred for 10 min and then ultrasonically dispersed for 15 min in a 2 mL vial. Subsequently, 0.71 g nematic liquid crystal 5CB was added to the above toluene solution, stirred for 10 min and then ultrasonically dispersed for 20 min. Finally, the mixture in the vial was held at 60 °C for 36 h to completely volatilize the solvent toluene.

### Characterization

TEM and high-resolution TEM were performed on JEM-2100Plus microscopes with acceleration voltages of 200 kV. The energy dispersive X-ray spectra (EDS) mapping and HAADF-STEM were collected on JEM-2100F and Talos F200X electron microscopes. The UV-vis-NIR spectrophotometer (Shimadzu 3700 DUV) was used for transmission and UV-vis spectra, Hitachi F-4700 fluorescence spectrophotometer for fluorescence spectra, and Hamamatsu (C11347) absolute PLQY spectrometer for photoluminescence quantum yield (365 nm, quartz cuvette). CD and CPL spectra were measured on a JASCO J-1500 and JASCO CPL-300 spectrophotometer, respectively. POM images were recorded on the material microscope upright Mshot MP41. The powder X-ray diffraction data (PXRD) was collected on Rigaku Smart Lab Diffractometer in the range of 20° to 70° (2θ) with Cu Kα radiation (wavelength: $\lambda = 1.54178$ Å).

## Data availability

The dataset supporting the findings of this study has been deposited in the Zenodo repository[65], and is available at https://doi.org/10.5281/zenodo.10258367. Source data are provided with this paper.

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

## Acknowledgements

T.Z., Y.Z., Y.W., S.Z., M.Zhang, G.L., Q.G., Z.T. and Z.L. were supported by the National Key Research and Development Program of China (grant 2021YFA1500400); the National Natural Science Foundation of China (grants 22071226, U1932213, 22271265, and 22101270); the Hundred Talent Program of the Chinese Academy of Sciences (grant KJ2060007002); the Collaborative Innovation Program of Hefei Science Center, Chinese Academy of Sciences (grant 2022HSC-CIP016); the Funding of University of Science and Technology of China (grants KY2060000168, YD2060002013, KY2060000198, and KY2060000235); and the Anhui Provincial Natural Science Foundation (grant BJ2060190120). The authors thank Dr. Yi Li and Dr. Liang Wu from the University of Science and Technology of China for helpful discussions, Prof. Mingming Ma's group from the University of Science and Technology of China for the CD test, Mr. Aoyuan Cheng from the University of Science and Technology of China for the PL decay test, Miss Qingqing Yan from the University of Science and Technology of China for the XRD test, Dr. Tanwei Li and Dr. Lei Shi at the University of Science and Technology of China for TEM test. The authors thank Dr. Huijuan Wang at Experimental Center of Engineering and Materials Sciences, University of Science and Technology of China for HADDF-STEM test. This work was partially carried out at the USTC Center for Micro and Nanoscale Research and Fabrication.

## Author contributions

T.Z. conceived the idea and supervised the project. T.Z., Y.Z. wrote the paper. Y.Z. carried out the experiments and analyzed the results. Y.W. and Y.S. helped to prepare figures. S.Z., M.Zhang, G.L., Q.G., Z.T., Z.L. and H.Y. helped to collect the data. S.J. and M.Zhu. helped to provide the CPL test. All authors discussed the results and assisted during manuscript preparation.

## Competing interests

The authors declare no competing interests.
