## [Peer Review File · Nature Communications]

Helical-caging enables single-emitted large asymmetric full-color circularly polarized luminescenceReviewers' comments:

Reviewer #1 (Remarks to the Author):

In this paper, the authors reported a new single-emitted WCPL system SWEG by combining whole-vis-spectrum emissive white quantum dots and chiral nematic liquid crystals. An impressive g_{lum} value of 0.8 is reached. The authors further achieved customized-color CPL, built a prototype WCPL device, and demonstrated its use in photoinduced enantioselective polymerization. The results should be interesting to the community of materials researchers. Some issues should be addressed before possible acceptance.

1. It is not suitable to call the coated LED as WLED. Usually, circularly polarized light emitting diodes (CP-LED) refer to electroluminescence, not photoluminescence (e.g., 10.1016/j.matt.2022.05.012; 10.1039/C9CS00680J; 10.1002/adma.202204253). Is it possible to develop WCPL-LED based on SWEG?
2. More details are needed to prove the WCPL-induced enantioselective polymerization of TDA. For example, controlled experiments should be performed to rule out other possibilities, such as samples illuminated under unpolarized light and under real CPL from commercially-available instrument, different substrates (quartz is a chiral material) and solvents, CD and CPL spectra of both TDA and PTDA, light power dependence, batch effect of samples.
3. It is important to take a time-resolved photoluminescence to gain the knowledge on emission life time of SWEG and the CPL emission mechanism.
4. Please explain why there are differences of CPL and g_{lum} values among SWEG samples with different doping ratios. Is there a relationship between the g_{lum} and doping ratio?
5. Fig 2f is the PL of SWEG, not "WCPL". Please show the whole visible range of the PL (400-800 nm) to proof the white-light emission. Similarly, the WCPL signals should be shown between 400-800 nm, not just 450-580 nm which is narrow.
6. Please check the caption of Fig 2:
i-g, CPL spectra (i) and corresponding symmetry factor (g_{lum}) spectra (g) of SWEG.

Reviewer #2 (Remarks to the Author):

In this manuscript entitled "Helical-caging enables single-emitted, large asymmetric white circularly polarized luminescence", the authors described a single-emitted white circularly polarized luminescence (WCPL) system with a maximum g_{lum} value of 0.8 by doping the whole-vis-spectrum emissive white quantum dots into chiral liquid crystals, further realizing customized-color CPL and WCPL-induced enantioselective polymerization. However, there are several major contradictions in the Manuscript far from considering publication:

- 1) Authors used different three chiral LCs (R/S811 ratio: 23%, 26%, and 29%) to realize red, green, and blue CPL, respectively. Obviously, they could not claim that they realized white CPL through single-emitted WCPL;

2) High PLQY (66%) was reported in Manuscript, but I did not see the PLQY of LC-doping system. Furthermore, the CPL claimed by authors in Fig. 3d was still a photo-excited CPL, rather than electroluminescence, which is quite different with previous reports on CPL (Adv. Mater., 2022, 34, 2109147; Angew. Chem. Int. Ed. 2022, 61, e2022022).

3) Actually, the WCPL-induced enantioselective polymerization mechanism should be further studied in Fig. 4. And whether the circularly polarized light generated by physical method ($g_{lum} = 2.0$) can be used to conduct this reaction?

More comments to the authors:

4) The Figure 1(ii, iii) exhibits the 5CB and chiral nematic liquid crystal with multicolor. Generally, the structural color of the liquid crystal is homogeneous at a specific helical pitch. The authors please revise the Figure to avoid misleading to audience.

5) The authors emphasize that the nematic liquid crystal 5CB is selected as an outstanding chiral host for its favorable compatibility, but lack of any evidence. In general, QDs perform poor compatibility with organic liquid crystal molecules. Moreover, the polarizing optical microscopy (POM) image in Figure 2e showed obvious defects in the LC texture, which also can illustrate the poor compatibility.

6) In customized-color CPL and device construction, the authors achieve strong mirror-image CPL signals with corresponding emissions by using different narrow bandpass filters. However, the range of photonic band in 5CB is around 50 nm in a fixed helical pitch, which is less than the whole visible regions. The authors please supply the g_{lum} value of 633, 540, and 455 nm in Figure 4f and the photonic band of 5CB. I think it will be small if the photonic band without overlapping the corresponding customized-color.

7) The large asymmetric white circularly polarized luminescence should be possessed a large g_{lum} in the whole visible regions, rather than in a specific region (Figure 2i). Moreover, the high $g_{lum} > 1.0$ in LC has been reported in the past few years (Laser Photonics Rev. 2022, 16, 2200115; Matter 2022, 5, 837-875; Macromol. Rapid Commun. 2021, 42, 2000548), so the $g_{lum} = 0.8$ is not enough large.

Reviewer #3 (Remarks to the Author):

In the manuscript, the authors report a spiral white emission generator composed of synthesized whole-vise-spectrum emissive QDs and chiral nematic LCs. White CPL is claimed with a maximum g_{lum} value of 0.8. However, it is not a real white CPL. The emission of QDs is white light, but the photonic band gap of LC is not. For example, the obtained CPL in Fig. 2i and 2j only covers 450 nm-580 nm. Although the photonic band gap can be adjusted by different amount of chiral dopant, each one is too narrow to cover whole-vise-spectrum to give a real white CPL. Therefore, "white circularly polarized luminescence" claimed in the whole manuscript is not accurate and this manuscript can be hardly recommended for publication in Nature Communications. Besides, there are some problems need to be solved before transferring to another journal.

1. The spiral white emission generator was obtained by mixing the QDs in LCs. As I know, the simple mixing method will face severe problem of phase separation. The authors should discuss how to avoid it.

2. Enantioselective polymerization of TDA was reported. CPL gives enantioselectivity. Here the effective wavelength should be discussed. Since the white CPL does not cover the whole-visible-spectrum. Which one is applied for this application? Is it the one shown in Fig. 2 with blue CPL? Which wavelength is more effective to give the enantioselectivity?
3. More WCPL references should be included, especially the ones based on LC system.

Reviewers' comments:

Reviewer #1 (Remarks to the Author):

In this paper, the authors reported a new single-emitted WCPL system SWEG by combining whole-vis-spectrum emissive white quantum dots and chiral nematic liquid crystals. An impressive g_{lum} value of 0.8 is reached. The authors further achieved customized-color CPL, built a prototype WCPL device, and demonstrated its use in photoinduced enantioselective polymerization. The results should be interesting to the community of materials researchers. Some issues should be addressed before possible acceptance.

We appreciate the Reviewer's helpful evaluation of the work and have now addressed all mentioned issues.

1. It is not suitable to call the coated LED as WLED. Usually, circularly polarized light emitting diodes (CP-LED) refer to electroluminescence, not photoluminescence (e.g., 10.1016/j.matt.2022.05.012; 10.1039/C9CS00680J; 10.1002/adma.202204253). Is it possible to develop WCPL-LED based on SWEG?

We have now changed the initial "WLED" to "coated CPL devices" of our previous devices in the revised manuscript, and further developed the electroluminescent CPL device as encouraged.

We first tuned the liquid crystal characteristic, using the freestanding liquid crystal polymer film instead of initial 5CB, and then combined with a designed multilayered white QLED to build the electroluminescent CPL device (Fig. R1a).

The white QLED (Fig. R1b), composed of ITO//PEDOT:PSS//PVK//QDs (Cu-Ga-S/ZnS)//MgZnO//Al, was attempted by all-solution processing¹. The collected EL spectrum and corresponding CIE color coordinates (Fig. R1c-d) confirmed the construction of the electroluminescent CPL device. Bright EL images of white QLED and the electroluminescent CPL device were shown in Fig. R1e-f. We also provided the g_{lum} spectra (Fig. R1g-i) of the free-standing liquid crystal films^{2,3} attached to the white QDs luminescent layer to evidence the electroluminescent CPL generation.

We have now included the electroluminescent CPL device as Fig. 3g-i in the revised manuscript and Supplementary Fig. 16 in the revised Supplementary Information.

Fig. R1 | Schematic illustration and characterizations of the electroluminescent CPL device. a, Schematic of the electroluminescent CPL device. **b,** Energy band diagram of the solution-processed, bottom-emitting multilayered white QLED. **c-d,** Collected EL spectrum (**c**) and the corresponding CIE color coordinates (**d**) of the electroluminescent CPL device. **e-f,** Bright EL images of the white QLED (**e**) and the electroluminescent CPL device (**f**). **g-i,** Mirror-imaged g_{lum} spectra of white QDs luminescent layer and liquid crystal films with different photonic bandgaps in blue (**e**), green (**h**) and red (**j**) region.

2. More details are needed to prove the WCPL-induced enantioselective polymerization of TDA. For example, controlled experiments should be performed to rule out other possibilities, such as samples illuminated under unpolarized light and under real CPL from commercially-available instrument, different substrates (quartz is a chiral material) and solvents, CD and CPL spectra of both TDA and PTDA, light power dependence, batch effect of samples.

As suggested, we returned to the lab and thoroughly studied the CPL-induced enantioselective polymerization of TDA under a series of parameters:

- Polarization of light:

The PTDA films showed no CD signal induced by unpolarized light while showed two opposite CD signals induced by CPL from left- and right-handed circular polarizers, respectively (Fig. R2).

Fig. R2 | **a**, CD spectrum of the PTDA film co-induced by unpolarized light and shortwave UV. **b**, CD spectra of the PTDA films co-induced by real CPL (left- and right-handed) from commercially-available instrument and shortwave UV.

- Different substrates:

Glass, PET and PS—the glass showed no CD signal while the PET and PS themselves showed noisy and uneven CD signals (Fig. R3).

With the Reviewer's notice, we have now clarified the "quartz" to "glass" in the revised manuscript.

Fig. R3 | **a**, CD spectrum of the glass substrate. **b-c**, CD spectra of the PET substrate. **d**, CD spectrum of the PTDA film on the PET substrate. **e-f**, CD spectra of the PS substrates. **g**, CD spectrum of the PTDA film on the PS substrate.

- Different solvents:

Mixture of ethanol and deionized water, toluene and DMSO—mixture of ethanol and deionized water led to the more uniform film on the glass (Fig. R4).

Fig. R4 | **a-b**, Schematic illustration (**a**) and images (**b**) of the TDA films with different solvents on glasses: mixture of ethanol and deionized water (left), toluene (middle) and DMSO (right).

- CD and CPL spectra of the TDA and PTDA:

CD spectra of the TDA and PTDA—TDA was achiral (Fig. R5a-b);

PL spectra of the TDA and PTDA (Fig. R5c-d) indicated that non-photoluminescent samples showed no CPL signals.

Fig. R5 | **a**, CD spectrum of the TDA monomer film. **b**, CD spectra of the PTDA polymer films. **c**-**d**, PL spectra of the TDA monomer (**c**) and the chiral PTDA (**d**).

- Light power dependence:

UV light intensity was tuned from strong to weak—the CD signal narrowed and red-shifted (Fig. R6).

Fig. R6 | **a-c**, CD spectra of the chiral PTDA films co-induced by right-handed CPL generated from our material and strong (**a**), moderate (**b**), weak (**c**) shortwave UV.

- Batch effect of samples:

The samples under the same condition showed fine repeatability (Fig. R7).

Fig. R7 | a-c, Different batches of the PTDA films co-induced by right-handed CPL generated from our system and shortwave UV.

We have now summarized the above results in Table. 1 and included them as Supplementary Figs. 20-26 and Supplementary Table 1 in the revised Supplementary Information.

Table. 1 | Characterization results of the enantioselective polymerization of TDA under different conditions.

Content	Variable	Result
Polarization of light	Unpolarized light	No CD signal
	Circularly polarized light	Opposite CD signals
	CPL from polarizers	Opposite CD signals
Substrate	Glass	Opposite CD signals
	PET	Noisy CD signals
	PS	Noisy CD signals
Solvent	Mixture of DI water and ethanol	Well-distributed film
	Toluene	Uneven surface
	DMSO	Uneven surface
CD spectrum	TDA monomer	No CD signal
	PTDA	Opposite CD signals
CPL spectrum	TDA monomer	No CPL signal
	PTDA	No CPL signal
UV light power	Weak	CD signals
	Moderate	Slight blue-shift CD signal
	Strong	Widened CD signals
Batch effect	1	Fine repeatability
	2	Fine repeatability
	3	Fine repeatability

3. It is important to take a time-resolved photoluminescence to gain the knowledge on emission life time of SWEQ and the CPL emission mechanism.

We took the time-resolved photoluminescence spectrum of the system and its emission life time is *ca.* 420.13 ns (Fig. R8a), shorter compared to pure white QDs (600.98 ns), owing to the Förster resonance energy transfer occurred in the host-guest system⁴.

We have now provided the CPL emission mechanism⁵ in Fig. R8b-c. The UV light first illuminates on the QDs after penetrating chiral liquid crystals (non-selective absorption of UV light). Then the QDs are excited and the formed photoluminescence passes

through the chiral liquid crystal host, during which CPL generates since the chiral liquid crystals could reflect the same handedness CPL and pass the opposite one. The role of the chiral liquid crystals in the entire process is similar to that of circular polarizers, while the blending of the luminescent component into the chiral host largely simplifies the complex optical path design, paving the way for fabricating CPL devices.

We have now included the schemes as Supplementary Fig. 10 in the revised Supplementary Information.

Fig. R8 | Time-resolved photoluminescence decay curve and the CPL emission mechanism. a, The monitored emission wavelength is 500 nm ($\lambda_{\text{ex}} = 365$ nm). **b-c,** Schematic illustration of the CPL emission (**b**) and its mechanism (**c**).

4. Please explain why there are differences of CPL and g_{lum} values among SWEG samples with different doping ratios. Is there a relationship between the g_{lum} and doping ratio?

We initially achieved a highest g_{lum} value of 0.8 when the photonic bandgap of chiral liquid crystal was tuned to blue region where the strongest emission of the WQDs located. Meanwhile, the g_{lum} values remained 10^{-1} in the whole-vis region by tunable

photon bandgaps thanks to the wide emission range of the single white emitters.

The photonic bandgap of chiral liquid crystals could be flexibly tuned by changing the ratio of chiral dopants. Thus, the emission of the emitters is usually located within the photonic bandgap to obtain CPL with larger g_{lum} values⁵⁻⁷.

5. **Fig 2f** is the PL of SWEG, not “WCPL”. Please show the whole visible range of the PL (400-800 nm) to proof the white-light emission. Similarly, the WCPL signals should be shown between 400-800 nm, not just 450-580 nm which is narrow.

The PL spectra of the WQDs and the system at the range of 400-800 nm were shown in Fig. R9.

Fig. R9 | **a-b**, PL spectra of the white QDs (**a**) and the system (**b**). The sharp peaks at 730 nm represent the frequency-doubled peak of the excited light (365 nm).

The initial CPL spectra was kind of narrow (Fig. R10a), and suggested by the Reviewer, we broadened the CPL spectrum to 400-800 nm based on the liquid crystal polymer for its promising broad weak transmission range due to focal conic alignment⁸.

We have now achieved broad photonic bandgap and CD signal (Fig. R10b-c) using the liquid crystal polymer, and by doping the WQDs, broad CPL signals (Fig. R10d) were obtained.

Fig. R10 | **a**, The initial CPL spectra. **b-c**, Transmission spectrum (**b**) and CD spectrum (**c**) of the liquid crystal polymer. **d**, The broadened CPL spectra of the liquid crystal polymer doped with the white QDs.

6. Please check the caption of **Fig 2**:

i-g, CPL spectra (*i*) and corresponding symmetry factor (*glum*) spectra (*g*) of SWEG.

We have now corrected the caption in the revised manuscript.

Reviewer #2 (Remarks to the Author):

In this manuscript entitled “Helical-caging enables single-emitted, large asymmetric white circularly polarized luminescence”, the authors described a single-emitted white circularly polarized luminescence (WCPL) system with a maximum glum value of 0.8 by doping the whole-vis-spectrum emissive white quantum dots into chiral liquid crystals, further realizing customized-color CPL and WCPL-induced enantioselective polymerization. However, there are several major contradictions in the Manuscript far from considering publication:

We thank the Reviewer’s feedback on the work and have now clarified all mentioned points.

1) Authors used different three chiral LCs (R/S811 ratio: 23%, 26%, and 29%) to realize red, green, and blue CPL, respectively. Obviously, they could not claim that they realized white CPL through single-emitted WCPL;

Different ratios of the chiral dopants and liquid crystals lead to varied photonic bandgaps⁵⁻⁷. As the reviewer mentioned, we obtained red, green and blue CPL in the single-emitted system since the photonic bandgap of 5CB is around 50 nm, which limited us to achieve broad-wavelength white CPL.

With the Reviewer’s inspiration, we tried to extend our horizons to chiral polymer systems owing to their broad weak transmission range (formed by focal conic alignment)⁸ to achieve wide photonic bandgap towards realizing white CPL.

We chose the liquid crystal monomer RM257 and chiral dopants R/S5011 to construct the chiral liquid crystal bulk, and then introduced spacer EDDT and crosslinker PETMP (Fig. R11a) to fabricate the liquid crystal polymer system. Undergoing the process of solvent evaporation and UV irradiation (Fig. R11b), as expected, we obtained the broad transmission and CD spectra (Fig. R11c-d), which brought inevitability for white CPL.

Finally, the broad CPL signals at 400-800 nm (Fig. R12) indicated that we achieved white CPL by doping white QDs into the liquid crystal polymer.

We also tuned the initial “white CPL” to “full-color CPL” in the revised manuscript.

Fig. R11 | **a**, Molecular structures of the main components of the liquid crystal polymer. **b**, Schematic process of the fabrication of the liquid crystal polymer. **c-d**, Transmission spectrum (**c**) and CD spectrum (**d**) of the liquid crystal polymer.

Fig. R12 | a-b, Broad CPL spectra, generated by combining the liquid crystal polymer with the white QDs.

2) High PLQY (66%) was reported in Manuscript, but I did not see the PLQY of LC-doping system. Furthermore, the CPLED claimed by authors in Fig. 3d was still a photo-excited CPL, rather than electroluminescence, which is quite different with previous reports on CPLED (*Adv. Mater.*, 2022, 34, 2109147; *Angew. Chem. Int. Ed.* 2022, 61, e2022022).

As suggested, we tested the PLQY of the LC-doping system and obtained a value of *ca.*11.5%—lowered compared to the parented QDs (due to the destruction of QD surface structure by the liquid crystal) but would not affect the practice in this work.

With the motivation of the Reviewer, we proposed an electroluminescent CPL device based on the white QDs (Cu-Ga-S/ZnS) and a freestanding chiral liquid crystal film^{2,3}. We went back to the lab and strived to put it into practice.

The white QD was applied as the single emitter to fabricate the white QLED through an all-solution process¹. Then the bottom-emitting multilayered electroluminescent CPL device structure of ITO//PEDOT:PSS//PVK//white QDs//MgZnO//Al with the chiral liquid crystal film and its energy band diagram were schematically depicted in Fig. R13a-b. The as-collected EL spectrum at 10 V in Fig. R13c and its corresponding CIE color coordinates (Fig. R13d) proved the construction of the electroluminescent CPL device. Subsequently, an additional freestanding chiral liquid crystal film (upgraded from 5CB to another liquid crystal monomer) was added to the ITO layer to generate electroluminescent CPL. Finally, we achieved bright EL and CP-EL emission as shown in Fig. R13f-g. We also provided the g_{lum} spectra of the chiral liquid crystal films with the QD layer in Fig. R13e to confirm their highly asymmetry for the

generation of CP-EL.

We have now changed the initial “CPLED” to “coated CPL devices” for the previous photoluminescent device and included the updated electroluminescent CPL device as Fig. 3g-i in the revised manuscript and Supplementary Fig. 16 in the revised Supplementary Information.

Fig. R13 | Schematic illustration and characterizations of the electroluminescent CPL device.

a, Schematic of the electroluminescent CPL device. **b**, Energy band diagram of a solution-processed, bottom-emitting multilayered white QLED. **c-d**, Collected EL spectrum (**c**) and the corresponding CIE color coordinates (**d**) of the electroluminescent CPL device. **e**, Mirror-imaged g_{lum} spectra of the white QDs luminescent layer and liquid crystal films with different photonic bandgaps. **f-g**, Bright EL images of the white QLED (**f**) and the electroluminescent CPL device (**g**).

3) Actually, the WCPL-induced enantioselective polymerization mechanism should be further studied in Fig. 4. And whether the circularly polarized light generated by physical method ($g_{lum} = 2.0$) can be sued to conduct this reaction?

The formation mechanism of chiral PTDA chains is now illustrated in Fig. R14⁹. The photo-initiation of polymerization reaction depends on the excitation of the TDA monomers in the shortwave UV region (< 310 nm), where the TDA monomers could

form an oligomer. After that, two ways are considered for the chain propagation. One is that the energy transfers from an excited monomer to the polymer chain triggered by shortwave UV. The other is to excite the oligomer chain directly by CPL when the duplicate units are more than 5. The thus-formed asymmetric carbenes might be aligned in a suitable orientation and reacted with the neighboring TDA monomers, eventually leading to the formation of the chiral PTDA chains directed by the handedness of CPL.

We used commercial circular polarizers to trigger the polymerization reaction of TDA monomers. We showed a photon-to-matter case, that is, employing our white QDs (Cu-Ga-S/ZnS) excited by 365 nm UV and commercial circular polarizers to generate CPL and then illuminated the light on the TDA monomer films assisted by shortwave UV (254 nm) for the chain propagation of the chiral PTDA (Fig. R15a).

After simultaneous explosion under left- or right-handed CPL and shortwave UV for 15 s, the virtually transparent TDA films turned to blue and the CD analysis of the samples showed obviously opposite Cotton effect (Fig. R15b), proving the formation of the chiral PTDA.

We also did a comprehensive study of the TDA polymerization—tuning the solvents, substrates and light power, to generalize this strategy. The schemes and results were included as Supplementary Fig. 17, Supplementary Figs. 20-26 and Supplementary Table. 1 in the revised Supplementary Information.

Fig. R14 | Schematic illustration of the mechanism of the enantioselective polymerization of TDA. Illustration of the formation mechanism of the chiral PTDA chains triggered by shortwave UV, CPL and CPL assisted by shortwave UV.

Fig. R15 | Characterizations of the enantioselective polymerization of TDA with commercial circular polarizers. **a**, Schematic drawing of the enantioselective polymerization of the TDA films co-induced by CPL generated by commercial circular polarizers and shortwave UV. **b**, CD spectra of the chiral PTDA induced by left- and right-handed CPL generated by commercial circular polarizers with the assistance of shortwave UV.

4) The Figure 1(ii, iii) exhibits the 5CB and chiral nematic liquid crystal with multicolor. Generally, the structural color of the liquid crystal is homogeneous at a specific helical pitch. The authors please revise the Figure to avoid misleading to audience.

We have now amended the color in the revised manuscript (Fig. 1) and also applied it to other related figures.

5) The authors emphasize that the nematic liquid crystal 5CB is selected as an outstanding chiral host for its favorable compatibility, but lack of any evidence. In general, QDs perform poor compatibility with organic liquid crystal molecules. Moreover, the polarizing optical microscopy (POM) image in Figure 2e showed obvious defects in the LC texture, which also can illustrate the poor compatibility.

We tried other representative liquid crystals (such as E7) in our previous attempts, and found that E7 could damage the surface structure of the QDs more seriously and finally caused the fluorescence quenching (Fig. R16a) compared to 5CB.

We admitted that the QDs perform imperfect compatibility with the liquid crystal molecules, while the serious aggregation and precipitation of QDs could be largely avoided macroscopically for more than 30 days in 5CB through ultrasonication and stirring (Fig. R16b) and the other results also indicated no significant impact on characterization and subsequent applications.

We have included the results as Supplementary Fig. 7 in the revised Supplementary Information.

Fig. R16 | Images and schematic illustration of the white QDs doped chiral liquid crystal. a, Images of the white QDs doped chiral liquid crystal E7 after 8 h. **b,** Images of the white QDs doped chiral liquid crystal 5CB under indoor light (left) and UV light (right) at day 0 and after 30 days with ultrasonication and stirring.

6) In customized-color CPL and device construction, the authors achieve strong mirror-image CPL signals with corresponding emissions by using different narrow bandpass filters. However, the range of photonic band in 5CB is around 50 nm in a fixed helical pitch, which is less than the whole visible regions. The authors please supply the g_{lum} value of 633, 540, and 455 nm in Figure 4f and the photonic band of 5CB. I think it will be small if the photonic band without overlapping the corresponding customized-color.

The photonic bandgaps of the chiral liquid crystals with different chiral dopant ratios were shown in Fig. R17a-c. The regulation of photonic bandgaps made the whole-wavelength CPL achieved. Whereafter, we could obtain customized-color CPL in the corresponding region, that is, customized-color CPL within the range of 520-800 nm could be achieved in the red region (Fig. R17d-f), satisfying specific CPL requirements while filtering out excess white light. As expected, the entire-visible-spectrum coverage of white QDs made the obtained g_{lum} values remained 10^{-1} regardless of wavelength (Fig. R17g-i).

We have now refined the initial descriptions and added more detailed schemes to avoid misunderstanding and included them as Supplementary Fig. 15 in the revised Supplementary Information.

Fig. R17 | Characterizations of the photonic bandgap and g_{lum} values. a-c, Transmission spectra of chiral liquid crystals with different photonic bandgaps in blue (a), green (b) and red (c) region, respectively. **d-f**, Schematic illustration of the generation of visualized customized-color CPL. **g-i**, g_{lum} spectra of customized-color CPL systems at 455 nm (g), 540 nm (h) and 633 nm (i).

7) The large asymmetric white circularly polarized luminescence should be possessed a large g_{lum} in the whole visible regions, rather than in a specific region (Figure 2i). Moreover, the high $g_{lum} > 1.0$ in LC has been reported in the past few years (Laser Photonics Rev. 2022, 16, 2200115; Matter 2022, 5, 837-875; Macromol. Rapid Commun. 2021, 42, 2000548), so the $g_{lum} = 0.8$ is not enough large.

With the encouragement of Reviewer, we have now achieved broad CPL in the whole visible region (shown in Fig. R11-12).

In our work, the g_{lum} value of 0.8 we obtained is sufficient for exploring the subsequent device construction and photo-induced polymerization.

Besides, as encouraged by the Reviewer, we are constantly hardworking in our lab for the amplification the g_{lum} values.

Reviewer #3 (Remarks to the Author):

In the manuscript, the authors report a spiral white emission generator composed of synthesized whole-vise-spectrum emissive QDs and chiral nematic LCs. White CPL is claimed with a maximum glum value of 0.8. However, it is not a real white CPL. The emission of QDs is white light, but the photonic band gap of LC is not. For example, the obtained CPL in Fig. 2i and 2j only covers 450 nm-580 nm. Although the photonic band gap can be adjusted by different amount of chiral dopant, each one is too narrow to cover whole-vise-spectrum to give a real white CPL. Therefore, “white circularly polarized luminescence” claimed in the whole manuscript is not accurate and this manuscript can be hardly recommended for publication in Nature Communications. Besides, there are some problems need to be solved before transferring to another journal.

We obtained red, green and blue CPL in the single-emitted system resulted from the varied photonic bandgaps by adjusting the doping ratio of the chiral dopants⁵⁻⁷. However, the narrow photonic bandgap of 5CB limited us to achieve broad-spectra white CPL.

The inherent photonic bandgap properties of chiral liquid crystals forced us to shift attention to other chiral systems—for example, the liquid crystal polymer, whose possibility for broad transmission (because of conic alignment)⁸.

We used the liquid crystal monomer RM257, chiral dopants R/S5011, spacer EDDT and crosslinker PETMP (Fig. R18a) to fabricate the liquid crystal polymer system since it could provide high asymmetry and the formation of polymers might change its inherent properties. After a process of solvent evaporation and UV irradiation (Fig. R18b), we obtained the broad transmission and CD spectra (Fig. R18c-d)—indicating that the liquid crystal could be a promising candidate for real white CPL.

As expected, we achieved the real white CPL by doping white QDs into the liquid crystal polymer (Fig. R19).

We also tuned the initial “white CPL” to “full-color CPL” in the revised manuscript as suggested.

Fig. R18 | **a**, Molecular structures of the main components of the liquid crystal polymer. **b**, Schematic process of the fabrication of the liquid crystal polymer. **c-d**, Transmission spectrum (**c**) and CD spectrum (**d**) of the liquid crystal polymer.

Fig. R19 | **a-b**, Broad CPL spectra, generated by doping white QDs into the liquid crystal polymer.

1. The spiral white emission generator was obtained by mixing the QDs in LCs. As I know, the simple mixing method will face severe problem of phase separation. The authors should discuss how to avoid it.

As for the problem of phase separation of 5CB and the white QDs, we took turns using ultrasonication and stirring during the solvent evaporation process, resulting in macroscopically favorable dispersion more than 30 days (Fig. R20a) for practice applications. We chose 5CB as the chiral host for its preferable compatibility and less destructiveness compared to some other representative liquid crystals (*e.g.*, E7) in our previous attempts, since E7 would more severely damage the surface structure of the white QDs (Fig. R20b), leading to fluorescence quenching and more prone to phase separation.

We have included the results as Supplementary Fig. 7 in the revised Supplementary Information.

Fig. R20 | Images and schematic illustrations of the white QDs doped chiral liquid crystal. a, Images of the white QDs doped chiral liquid crystal 5CB under indoor light (left) and UV light (right) at day 0 and after 30 days with ultrasonication and stirring. **b,** Images of the white QDs doped chiral liquid crystal E7 after 8 h.

2. Enantioselective polymerization of TDA was reported. CPL gives enantioselectivity. Here the effective wavelength should be discussed. Since the white CPL does not cover the whole-visible-spectrum. Which one is applied for this application? Is it the one shown in Fig. 2 with blue CPL? Which wavelength is more effective to give the enantioselectivity?

Blue CPL was applied for the enantioselective polymerization of TDA in the initial

manuscript. Further, we employed red, green and blue CPL by regulating the photonic bandgap of the system and shortwave UV to co-induce the polymerization of TDA, and then we found that there was no obvious difference in the CD signals of the chiral PTDA (Fig. R21).

We also studied a series of other parameters including substrates, solvents, UV light intensity and light polarization, and have now included them as Supplementary Figs. 20-26 in the revised Supplementary Information.

Fig. R21 | Characterizations of the enantioselective polymerization of TDA (taking right-handed CPL as the example). a-c, CD spectra of the chiral PTDA induced by right-handed red (a), green (b) and blue (c) CPL generated by the system (with regulated photonic bandgaps) with the assistance of shortwave UV.

3. More WCPL references should be included, especially the ones based on LC system.

We strived to find more references and have now included them as references 15-17 and 20-45 in the revised manuscript.

Reference:

1. Kim, J. H. *et al.* White electroluminescent lighting device based on a single quantum dot emitter. *Adv. Mater.* **28**, 5093–5098 (2016).
2. Han, H. *et al.* High-performance circularly polarized light-sensing near-infrared organic phototransistors for optoelectronic cryptographic primitives. *Adv. Funct. Mater.* **30**, 2006236 (2020).
3. Jeong, S.M. *et al.* Highly circularly polarized electroluminescence from organic light-emitting diodes with wide-band reflective polymeric cholesteric liquid crystal films. *Appl.*

- Phys. Lett.* **90**, 211106 (2007).
4. Xu, L. *et al.* Efficient circularly polarized electroluminescence from achiral luminescent materials. *Angew. Chem. Int. Ed.* **62**, e202300492 (2023).
 5. Yang, X., Zhou, M., Wang, Y. & Duan, P. Electric-field-regulated energy transfer in chiral liquid crystals for enhancing upconverted circularly polarized luminescence through steering the photonic bandgap. *Adv. Mater.* **32**, e2000820 (2020).
 6. Xu, M. *et al.* Assembling semiconductor quantum dots in hierarchical photonic cellulose nanocrystal films: circularly polarized luminescent nanomaterials as optical coding labels. *J. Mater. Chem. C* **7**, 13794–13802 (2019).
 7. Guo, Q. *et al.* Multimodal-responsive circularly polarized luminescence security materials. *J. Am. Chem. Soc.* **145**, 4246–4253 (2023).
 8. Zhang, P., Zhou, G., Haan, L.T. & Schenning, A.P.H.J. 4D chiral photonic actuators with switchable hyper-reflectivity. *Adv. Funct. Mater.* **31**, 2007887 (2021).
 9. Wang, X. *et al.* Circularly polarized light source from self-assembled hybrid nanoarchitecture. *Adv. Opt. Mater.* **10**, 2200761 (2022).

REVIEWER COMMENTS

Reviewer #1 (Remarks to the Author):

The manuscript was carefully revised. All of my concerns were well solved. I think it can be accepted in the present format.

Reviewer #2 (Remarks to the Author):

Although the g_{lum} of around 0.8 is not so impressive, the fact that the helical-caging single-emitter achieves large asymmetric full-color circularly polarized luminescence is amazing. Since the revision has been made clear several points, like full-color instead of white color, and the working mechanism of the photoinduced enantioselective polymerization, now it can be publishable in Nature Communications.

Two very relative latest references should be included: J. Am. Chem. Soc., 2022, 144, 20773-20784; J. Am. Chem. Soc., 2023, 145, 12951-12966.

Reviewer #3 (Remarks to the Author):

In the manuscript, the authors reported a full-color CPL, composed of synthesized white emissive QDs and chiral nematic liquid crystals. A maximum g_{lum} value of 0.8 was obtained. The authors also expanded the SFEG for future applications of electroluminescent CPL devices and photoinduced enantioselective polymerization of diacetylene derivatives. The results are interesting. However, some issues should be addressed before possible acceptance.

1. Strong mirror-image CPL signals with the g_{lum} value up to 0.8 and remains 10⁻¹ in the whole-vis region are claimed by the authors. However, this is not true. In Fig. 2, the CPL and g_{lum} spectra go across 0 and even become opposite at the edge. g_{lum} value of 10⁻¹ in the whole-vis region cannot be claimed, and proper discussion of the opposite value should be included.
2. Supplementary Fig. 8 should be put into Fig. 2h to compare with g_{lum} spectra. In line 103-105, the authors claimed "located at the edge of the photonic bandgap, the emission of the embedded emitter allows for enhancement." Do the results support this statement? Detailed discussion should be given. In addition, labels on the y axis of Fig. 2g should be marked.
3. The g_{lum} spectra should be given for the liquid crystal polymer system, which are crucial for evaluating CPL.
4. Many experimental data are put in the supplementary material. Most of them should be discussed instead of just listed in the manuscript. In addition, a lot of qualitative descriptions are presented and need detailed information. For example, supplementary Fig. 24 claims that the chirality of PTDA films depends on the intensity of UV. Quantitative description should be given instead of using the words "strong, moderate and weak". And the quantitative description, such as wavelength, intensity of CPL and UV, should be given for the enantioselectivity experiments in the manuscript.
5. No labels on the y axes in Supplementary Fig. 13d, Fig. 14, Fig. 20-26. The values are important and needed to evaluate the chirality.
6. Different colors are presented without any discussion in Supplementary Fig. 27 and Fig. 28. Detailed description and information should be given.

Reviewers' comments:

Reviewer #1 (Remarks to the Author):

The manuscript was carefully revised. All of my concerns were well solved. I think it can be accepted in the present format.

We are grateful for the Reviewer's examination.

Reviewer #2 (Remarks to the Author):

Although the glum of around 0.8 is not so impressive, the fact that the helical-caging single-emitter achieves large asymmetric full-color circularly polarized luminescence is amazing. Since the revision has been made clear several points, like full-color instead of white color, and the working mechanism of the photoinduced enantioselective polymerization, now it can be publishable in Nature Communications.

Two very relative latest references should be included: J. Am. Chem. Soc., 2022, 144, 20773-20784; J. Am. Chem. Soc., 2023, 145, 12951-12966.

We thank the Reviewer for the feedback on the work and have now included the mentioned papers as refs 51 and 4 in the revised manuscript.

Reviewer #3 (Remarks to the Author):

In the manuscript, the authors reported a full-color CPL, composed of synthesized white emissive QDs and chiral nematic liquid crystals. A maximum g_{lum} value of 0.8 was obtained. The authors also expanded the SFEG for future applications of electroluminescent CPL devices and photoinduced enantioselective polymerization of diacetylene derivatives. The results are interesting. However, some issues should be addressed before possible acceptance.

We appreciate the Reviewer's evaluation of the work and have now clarified all mentioned concerns.

1. Strong mirror-image CPL signals with the g_{lum} value up to 0.8 and remains 10^{-1} in the whole-vis region are claimed by the authors. However, this is not true. In Fig. 2, the CPL and g_{lum} spectra go across 0 and even become opposite at the edge. g_{lum} value of 10^{-1} in the whole-vis region cannot be claimed, and proper discussion of the opposite value should be included.

In our work, 10^{-1} order g_{lum} values in the visible region (Fig. R1) could be obtained by different samples with tunable photonic bandgaps of the liquid crystal. A single one was not competent, as the Reviewer mentioned.

We have now refined the statements in the revised manuscript to avoid misunderstanding.

Fig. R1 | a-b, CPL and g_{lum} spectra of the SFEG with regulated photonic bandgaps.

The chiral dopant S811 could induce the nematic liquid crystal to left handedness, while R811 resulted in the right one. We obtained the opposite CPL signal to the handedness of the parented liquid crystal due to its selective reflection when the emission of the

embedded emitter was located in the photonic bandgap^{1,2}. However, since the oscillations of the density of states in the vicinity of the photonic bandgap^{3,4}, the weak reverse CPL signal could appear at the edge (such phenomena were observed in some similar systems⁵⁻⁷).

We have now discussed this phenomenon in the revised manuscript (lines 144-146).

2. Supplementary Fig. 8 should be put into Fig. 2h to compare with g_{lum} spectra. In line 103-105, the authors claimed “located at the edge of the photonic bandgap, the emission of the embedded emitter allows for enhancement.” Do the results support this statement? Detailed discussion should be given. In addition, labels on the y axis of Fig. 2g should be marked.

We have put the initial Supplementary Fig. 8 into Fig. 2 as the current Fig. 2g in the revised manuscript.

In chiral luminous systems based on liquid crystals, the long dwell time of the emitted photons leads to emission enhancement at the band edge^{3,8,9}. We initially introduced this statement to provide a relatively comprehensive description of chiral nematic liquid crystals; however, we did not conduct specific research on this property (that is, QD-emission located at the band edge would lead to the sacrifice of the g_{lum} value) since the QDs with broad-spectrum luminescence were used in our system. We thus have now removed such sentence in the revised manuscript to avoid confusion.

We have marked the y axis of the CPL spectra (the initial Fig. 2g, now Fig. 2h) in the revised manuscript.

3. The g_{lum} spectra should be given for the liquid crystal polymer system, which are crucial for evaluating CPL.

With the Reviewer’s reminder, we further examined the g_{lum} values of the liquid crystal polymer system (Fig. R2). Compared to the SFEG, the liquid crystal polymer-based CPL architecture broadened the CPL spectra while causing some loss of g_{lum} values.

Fig. R2 | a-b, The corresponding g_{lum} spectra of the liquid crystal polymer system.

We have now included the g_{lum} spectra in the revised Supplementary Fig. 13c-d.

4. Many experimental data are put in the supplementary material. Most of them should be discussed instead of just listed in the manuscript. In addition, a lot of qualitative descriptions are presented and need detailed information. For example, supplementary Fig. 24 claims that the chirality of PTDA films depends on the intensity of UV. Quantitative description should be given instead of using the words “strong, moderate and weak”. And the quantitative description, such as wavelength, intensity of CPL and UV, should be given for the enantioselectivity experiments in the manuscript.

As suggested, we have now provided more discussions for supplementary data.

We have also added quantitative descriptions and information in the revised Supplementary Figs. 18-24.

We controlled the light intensity generated by the UV lamp (254 nm, 16 W) via adjusting the distance (from 5 to 15 cm) between the sample and lamp. The corresponding schematic illustration was shown in Fig. R3 and the quantitative description was included in the figure caption.

Fig. R3 | **a**, Schematic illustration of the light intensity generated by the UV lamp (254 nm, 16 W), regulated via the distance between the sample and light source. **b-d**, Corresponding CD spectra of the PTDA films with tuning sample-light distances, 15 cm (**b**), 10 cm (**c**) and 5 cm (**d**). With decreased distance, the polymerization became faster and more difficult to control and the corresponding CD spectra showed blueshifts and broadening. We chose the distance of 15 cm to carry out the enantioselectivity experiments.

For most enantioselectivity experiments, the CPL (luminescence peak at 500 nm, $|g_{lum}|$ of 0.6) was employed and the used UV lamp (254 nm, 16 W) was placed at a distance of 15 cm to the samples. Other contrast experiments with specific parameters were marked in the revised corresponding captions.

5. No labels on the y axes in Supplementary Fig. 13d, Fig. 14, Fig. 20-26. The values are important and needed to evaluate the chirality.

We have now marked all mentioned y axes.

6. Different colors are presented without any discussion in Supplementary Fig. 27 and Fig. 28. Detailed description and information should be given.

We have now provided detailed descriptions and information in the captions of the

revised Supplementary Fig. 25 and Fig. 26 (the initial Supplementary Fig. 27 and Fig. 28).

The caption of Supplementary Fig. 25: “Information encryption and anti-counterfeiting based on the SFEG. **a-c**, The pattern formed by the SFEG (with 26 wt% chiral dopants S/R811) showed green color at *ca.* 16 °C and provided information (color unit) change under natural light (**a**), right- (**b**) and left-handed circular polarizers (**c**), respectively. **d-f**, The pattern color changed to red when the temperature decreased, as well as giving distinctive information observed by different polarizers.”

The caption of Supplementary Fig. 26: “Information responses with different stimuli in the SFEG system (with different weight ratios of the chiral dopants). **a**, The color-unit-composed pattern became transparent at a high environmental temperature (here, it was 60 °C). **b-c**, The pattern changed from red (**b**) to mixed color (**c**) mediated by temperature decrease. **d**, Bright-white color emission achieved by 365 nm excitation.”

Reference:

1. Liu, S. *et al.* Circularly polarized perovskite luminescence with dissymmetry factor up to 1.9 by soft helix bilayer device. *Matter* **5**, 2319–2333 (2022).
2. Yang, X., Jin, X., Zheng, A. & Duan, P. Dual band-edge enhancing overall performance of upconverted near-infrared circularly polarized luminescence for anticounterfeiting. *ACS Nano* **17**, 2661–2668 (2023).
3. Schmidtke, J. & Stille, W. Fluorescence of a dye-doped cholesteric liquid crystal film in the region of the stop band: theory and experiment. *Eur. Phys. J. B* **31**, 179–194 (2003).
4. Xu, M. *et al.* Designing hybrid chiral photonic films with circularly polarized room-temperature phosphorescence. *ACS Nano* **14**, 11130–11139 (2020).
5. Han, D. *et al.* Sequentially amplified circularly polarized ultraviolet luminescence for enantioselective photopolymerization. *Nat. Commun.* **11**, 5659 (2020).
6. Lin, S. *et al.* Photo-triggered full-color circularly polarized luminescence based on photonic capsules for multilevel information encryption. *Nat. Commun.* **14**, 3005 (2023).
7. Xu, M. *et al.* Designing hybrid chiral photonic films with circularly polarized room-temperature phosphorescence. *ACS Nano* **14**, 11130–11139 (2020).
8. Yang, X., Jin, X., Zheng, A. & Duan, P. Dual band-edge enhancing overall performance of upconverted near-infrared circularly polarized luminescence for anticounterfeiting. *ACS*

Nano **17**, 2661–2668 (2023).

9. Kopp, V.I., Fan, B., Vithana, H.K.M. & Genack, A.Z. Low-threshold lasing at the edge of a photonic stop band in cholesteric liquid crystals. *Opt. Lett.* **23**, 1707–1709 (1998).

REVIEWERS' COMMENTS

Reviewer #3 (Remarks to the Author):

The manuscript was carefully revised. All of my concerns were well solved. I think it can be accepted in the present format in Nature Communications.